# Hormetic and Mitochondria-Related Mechanisms of Antioxidant Action of Phytochemicals

**DOI:** 10.3390/antiox8090373

**Published:** 2019-09-04

**Authors:** Rafael Franco, Gemma Navarro, Eva Martínez-Pinilla

**Affiliations:** 1Chemistry School, University of Barcelona, 08028 Barcelona, Spain; 2Centro de Investigación Biomédica en Red Enfermedades Neurodegenerativas (CiberNed), Instituto de Salud Carlos III, 28031 Madrid, Spain; 3Department of Biochemistry and Physiology, Faculty of Pharmacy and Food Sciences, University of Barcelona, 02028 Barcelona, Spain; 4Departamento de Morfología y Biología Celular, Facultad de Medicina, Universidad de Oviedo, 33006 Oviedo, Spain; 5Instituto de Neurociencias del Principado de Asturias (INEUROPA), 33003 Oviedo, Spain; 6Instituto de Investigación Sanitaria del Principado de Asturias (ISPA), 33011 Oviedo, Spain

**Keywords:** CNS, fava beans, glucose, fructose, oxidative stress, vitagenes

## Abstract

Antioxidant action to afford a health benefit or increased well-being may not be directly exerted by quick reduction-oxidation (REDOX) reactions between the antioxidant and the pro-oxidant molecules in a living being. Furthermore, not all flavonoids or polyphenols derived from plants are beneficial. This paper aims at discussing the variety of mechanisms underlying the so-called “antioxidant” action. Apart from antioxidant direct mechanisms, indirect ones consisting of fueling and boosting innate detox routes should be considered. One of them, hormesis, involves upregulating enzymes that are needed in innate detox pathways and/or regulating the transcription of the so-called vitagenes. Moreover, there is evidence that some plant-derived compounds may have a direct role in events taking place in mitochondria, which is an organelle prone to oxidative stress if electron transport is faulty. Insights into the potential of molecules able to enter into the electron transport chain would require the determination of their reduction potential. Additionally, it is advisable to know both the oxidized and the reduced structures for each antioxidant candidate. These mechanisms and their related technical developments should help nutraceutical industry to select candidates that are efficacious in physiological conditions to prevent diseases or increase human health.

## 1. Chemical Basis of Antioxidant Action

Any reduction-oxidation (REDOX) reaction follows well-established chemical laws. Almost any substance can be oxidized or reduced and this depends on a second reagent. In fact, a REDOX reaction requires two semi-reactions and two molecules: one that is oxidized (the reductant) and another that is reduced (the oxidant). Accordingly, any antioxidant molecule is a reductant able to reduce an oxidized reagent. The chemical laws guiding REDOX processes are valid in vivo and in vitro (i.e., in a test tube). As commented in a previous paper, the canonical function of antioxidants in the food industry is to increase the useful life of processed foods. In the case of animal-derived products, antioxidants are added to avoid/delay rotting (i.e., their action is to prevent putrefaction of dead matter) [1].

Nowadays, “antioxidants” is a word also used to refer to substances that provide benefit to humans due to REDOX-related capabilities. Importantly, it is often assumed that these capabilities are directly exerted on a given tissue or a given oxidative stressor. However, antioxidants may indirectly exert their beneficial effect. This paper aims at highlighting the hormetic and mitochondria-related mechanisms of antioxidants action that can be changed by the phytochemical use. 

## 2. Are Antioxidants Needed for Human Life?

The use of phytochemicals dates back to ancestral times. Paleontological and anthropological evidence demonstrates that Neanderthals, who until quite recently were regarded predominantly as meat-eaters, included plants in their diet. Valuous studies aimed at identifying entrapped material in calcified dental plaque show, quite surprisingly, that a Neanderthal group of hominids who inhabited El Sidrón cave (Asturias, Spain) 49,000 years ago, did not consume foods of animal origin but moved in the direction of a more “healthy” vegetable/plant-based diet, including mushrooms, pine nuts, and moss [2,3]. Apparently, our ancestors already noticed that some plants helped, not only calorically, but also in terms of well-being. More prevalent in higher primates, this “self-medication” is not exclusive to them but it is widespread within the animal kingdom [4,5,6]. 

A newborn does not need any plant derived antioxidant for living. If the child takes milk from the mother it may be hypothesized that mother’s milk already had its own antioxidants some of which could derive from ingested plants. But the use of formula milks and the lack of increased health problems in children raised with artificial milk prove that antioxidants are not required for a child’s healthy life. Commercially available infant milk contains plant derived products but few (or none) that are considered as “antioxidants”. On the one hand, they contain sugars and fatty acids/lipids, which are oxidants, in other words very reduced molecules that are oxidized in infant’s cells. On the other hand, they contain vitamins and no added antioxidant apart from, eventually, one acting as preservative. As we have previously argued, vitamins may have in vitro antioxidant action, but none in the basis of chemical rules, so they do not act as in vivo antioxidants [1]. According to Mayo Clinic the daily requirement of vitamin A for adult men is 900 µg (i.e., very low to have any overall direct antioxidant effect).

Fruits are among the first plant-derived foods introduced in child’s nutrition. Indeed, fruit has antioxidants but is the antioxidant content the reason for such early introduction of fruits in human diet? The answer is, likely not. Another relevant question is whether sugars (from fruits) can be the only sugar energy source in humans. Despite glucose and fructose in human nutrition derive from vegetables, the answer is that human metabolism has evolved to use glucose instead of fructose as main energy source. 

But are glucose or fructose antioxidants? From a chemical point of view, glucose is a reducing sugar whereas fructose is a non-reducing sugar. This nomenclature goes back to the first reactions using biological compounds that were designed to identify sugars in a mixture or in blood. Reduction is a relative term since it depends on the properties of a second component that may be reduced or oxidized. The classical technique to measure the reducing potential was to observe whether Cu^2+^ could be reduced to Cu^+^ in basic conditions of pH. In mild circumstances, this technique results in glucose being a reducing sugar (an aldose) and fructose (a ketose) being a non-reducing sugar. But if conditions are forced, fructose, due to ketose-aldose tautomerism may lead to the Cu^2+^/Cu^+^ reduction (i.e., fructose could react as a reducing sugar). Very importantly, these in vitro assays are of little usefulness in physiological conditions. In fact, may a reducing sugar (glucose) or a non-reducing sugar (fructose) be in vivo antioxidants? None of two sugars acts as antioxidants in humans [7,8,9], both are oxidized to obtain energy and, in the case of glucose, to (also) obtain reducing power to be used in innate antioxidant mechanisms. In summary, the non-reducing sugar, fructose, and the reducing sugar, glucose, are not in vivo-acting antioxidants, but the latter is required to provide the molecules needed by innate antioxidant human systems (see [1] for further details). 

## 3. When Does Antioxidant Intake Become Beneficial in Human Life?

Some evidence suggests that a high fructose dietary intake, in the form of cultivated fruits or sweeteners (sucrose or high fructose corn syrup), is related to the development of a variety of metabolic diseases [10,11]. Fructose absorbed in the gut is completely converted into triose-phosphate by fructokinase, aldolase B and triokinase, in the sequential steps of a metabolic pathway which is not controlled through feedback inhibition (ADP or citrate) [12]. Resulting metabolites may be oxidized, and converted into glucose and lactate to be released into the bloodstream, or converted into hepatic glycogen or triacylglycerol in liver cells by de novo lipogenesis, in an insulin-independent process. All together leads to increases in plasma triglycerides, insulin resistance, or high blood pressure [10,13,14]. Although the data about the adverse metabolic effects of fructose in humans are controversial [14], rodents fed with a high-fructose diet show fatty liver, impaired insulin sensitivity, or dyslipidemia [13,15,16,17], also affecting (negatively) the antioxidant status [8,9]. Similar results were reported in *Macaca fascicularis* and rhesus monkeys exposed to high fructose over a long period of time [18,19] (Table 1). 

Current human dietary habits are inherited from our ancient hominid specimens and are similar to that of wild monkeys and apes. How is it possible that dietary habits may cause chronic illnesses and health problems, mainly in Western societies? One answer comes from tackling the changes in food staples and food-processing procedures. In fact, some studies suggest that many harvested fruits and vegetables eaten by humans differ from the wild versions in regards to fatty acid level, macronutrients and phytochemical composition or fiber content. It is even possible that the use of food additives and supplements have detrimental effects. Let us take “designed fruits” as an example. These fruits are optimized by seed selection and cross-fertilization to have succulent pulps with dew or no seeds, thus becoming more attractive for the consumers. Surely, the most important difference lies on sweetness and, consequently, on the sugar content. Whereas wild fruits are rich in hexoses, as glucose and fructose, cultivated fruits have been genetically modified to be higher in sucrose, a glucose and fructose disaccharide for which our metabolism is not properly adapted [4,5]. In the seminal work of Schwitzer et al., (2008) we can find an exhaustive comparison between wild and cultivated fruits and vegetables in terms of nutrients and energy content. For example, the major sugars in harvested figs are 0.40% sucrose, 25.5% glucose and 23.40% fructose, in contrast to the pattern of the wild variety *Ficus insipida* that shows a 0.4% of sucrose, a 0.6% of glucose and a 0.3% of fructose [20]. In this sense, other studies have also identified lower monosaccharides/disaccharides ratios in modified fruits and vegetables [31,32,33]. Compelling evidence in recent years, has shown that, apart from the fructose content, the processing operations of commercial forms of fruit and vegetable food products influence the levels of a myriad of dietary phytochemicals [34]. In addition, additive and/or synergistic role of some flavonols present in culinary plants has been demonstrated and among them, myricetin, fisetin, quercetin, catechin and curcumin seem to inhibit fructose gut transport by glucose transporter 2 (GLUT2) and 5 (GLUT5), as shown in *Xenopus laevis* oocytes and in human intestinal Caco-2 cells [35,36]. 

When this nutritional information is added to physiological burden due to wear and/or malfunction occurring in aging or disease, humans need help. According to our view, the need of “so-called” antioxidants begins upon aging (i.e., when innate antioxidant mechanisms start to have some dysfunctions and alter well-being). Then, the search for interventions to keep healthiness is of vital importance and, in this sense, “antioxidants” have a huge potential. However, in our opinion, to take optimal profit of antioxidant-related interventions, more knowledge on mechanisms is needed.

## 4. Direct Mechanisms of Plant Antioxidant/Nutraceutical Action

A REDOX reaction may immediately take place when a reductant and an antioxidant meet. The kinetics coordinate is very important as there are some rules to fulfil to delay antioxidant actions. The naked eye is able to see quick oxidation when fruits, for instance bananas and apples, are peeled off. Instantaneously some component(s) of the fruit (reductant(s)), reacts with atmospheric oxygen (oxidant) and the process is visible by the appearance of a brown color. As the process is relatively quick, similar reactions take place when a human eats bananas or apples but locally (i.e., along the gastrointestinal tract). It is very unlikely that reductants remain intact until reaching a given tissue and reacting with “the undesirable” oxidant. In summary, ready-to-use reduction potential likely occurs locally [1]; in the case of food, thus potential is limited to the proximal structures of the gastrointestinal tract: esophagus and stomach (Figure 1).

Surely, REDOX reactions may be “delayed” if time of reaction is the limiting factor. The most common way to increase reaction rate in living organisms is by the action of bio-catalysts (i.e., of enzymes). In such cases, the reductant present in a plant-derived product may be considered as a drug orally taken which reaches the blood, is distributed and acts in the targeted tissue by, among other, activating a receptor or inhibiting an enzyme. In the case of a reductant able to act like a drug, it would reach the targeted tissue and react with a specific (undesirable) oxidant. In summary, this mechanism requires a reductant that travels to the target tissue(s) to meet a specific oxidant and a specific enzyme able to catalyze the REDOX reaction (Figure 1). Somewhat expected, such reductant has not yet been described (to our knowledge). A reductant acting by this mechanism would be a nutraceutical—“a type of food substance that helps to maintain health and prevent illness”—according with the Encyclopedia Britannica. Antioxidants research is a relatively young field and the tendency is to group molecules: polyphenols, flavonoids, etc. However, each molecule is different and it is likely that there are “good” and “bad” polyphenols. Identifying the “good” ones and demonstrating that there is a specific mechanism able to inactivate a noxious oxidant in a given tissue is an attractive possibility, even a necessity. In one of the most recent studies, toxicity of polyphenols or polyphenol-rich plant extracts has been addressed by quantitating and comparing the effects on survival in hepatic (HepG2), fibroblast (3T3), epithelial (A549 and Caco-2), and endothelial (HMEC-1) human cell lines. As a conclusion a list of the five most toxic and the five least toxic polyphenol-rich compounds was provided [37]. As pure molecules, naringin was the less toxic and kaempferol the most toxic. Accordingly, in the data sheet of kaempferol from Cayman chemical company indicates “acute toxicity and germ cell mutagenicity” (in humans); in addition, the product is “H301: toxic if swallowed” and “H341: suspect of causing genetic defects” [38]. It is important to highlight that any nutraceutical must be tested for genotoxicity to get approval to be released into the market.

From both human health and industrial point of views, another relevant aspect should not be forgotten which is bioavailability and the possibility that per-oral consumed molecules are transformed upon in vivo metabolism or gut fermentation/biotransformation [39]. On the one hand, bioaccessibility of the administered compound and its metabolites may vary depending on the “galenic” formulation of the phytochemical-containing product. One example is provided by comparing creams with microencapsulated phenolic acids and flavanols [40]; another is to find the best vehicle to increase the bioavailability of curcumin incorporated to bread [41]. On the other hand, total antioxidant capacity, measured for instance in blood, may be a convenient tool to quickly decide whether to continue or not the research and development of a substance or of an extract [42]. Nevertheless, caution is needed as total antioxidant capacity is not reflecting the whole potential of substances indirectly reinforcing antioxidant mechanisms. 

## 5. Hormetic and Replenishment Mechanisms of Indirect Antioxidant/Nutraceutical Action

Faulty oxidation in cells results in oxidative stress, which is harmful [43]. For instance, the electron transport and oxidative phosphorylation events taking place in the mitochondria are self-regulated; if cells have all the required components and the mitochondria are healthy no undesired REDOX-related effect is expected to occur. Oxidative stress appears as a threat to cell/organism survival when a component is reduced or the mitochondria do not properly work. 

A well-known indirect antioxidant strategy is to provide components (or precursors) of REDOX-related routes. One option is to supply molecules that participate in events whose dysfunction leads to oxidative stress. A second option is providing molecules that participate in innate mechanisms of detoxification. A few examples of those strategies are provided below.

A very successful way to use plant antioxidants is by increasing the concentration of endogenous substances that participate in REDOX mechanisms. A complete list of antioxidant supplements and its “galenic” formulations is out of the scope of the present paper; therefore, we will provide some examples of antioxidants derived from plants that are quite fruitful. For instance, supplement of coenzyme Q10, ubiquinol (ubiquinone in oxidized form), may replenish the compound if there is a shortage due to aging, disease or, eventually, malnutrition. The coenzyme Q10, essential for electron chain transport in mitochondria, can be found in a variety of vegetable sources: oranges, spinach, broccoli, soybeans, nuts, sesame seeds, etc. A recent report has confirmed shortage of coenzyme Q10 in centenarians and a correlation with increased oxidative stress [21]. Other benefits of supplementation with ubiquinol/ubiquinone include improvement in orthostatic hypotension [22], and prevention of renal alterations in a model of type II diabetes [23]. Interestingly, a recent double-blind placebo-controlled clinical trial has shown that ubiquinol is better than ubiquinone to increase the total levels of coenzyme Q10 [24] (Table 1).

Lipoic (or thioctic) acid is another endogenous component that may lead to oxidative stress if cell concentrations decay. It is found in a variety of plant-derived products: potatoes, spinach, broccoli, carrots, tomatoes, rice bran, etc. In mammalians, lipoic acid participates in at least five different enzymatic systems. For example, two of the enzymes of the Kreb’s cycle, which occurs in mitochondria to produce reducing power in the form of reducing nicotine adenine dinucleotide (NADH) and reduced flavin-adenine dinucleotide (FADH_2_), need lipoic acid. Benefits of lipoic acid supplementation are less evident than those attributed to coenzyme Q10 but it is thought that they may be helpful to combat oxidative stress (by mechanisms not yet deciphered). A systematic review and meta-analysis indicates that these benefits include improvement of biomarkers of diabetes and of inflammation [25] (Table 1). 

Whereas supplements of lipoic acid in diets are obtained from plant-derived products, coenzyme Q10 used in the food or cosmetic industry is synthetic or produced by microbes in fermenters. The latter leads to a low yield and is relatively expensive. Plant-derived raw materials or products are cost-effective alternatives. Production of coenzyme Q10 from plants may be a relatively cheap but environment-friendly approach that is explored by the supplement’s industry [44].

Probably, one of the main antioxidant mechanisms exerted by plant components gets unnoticed. It is indirect and counterintuitive and consists of boosting the innate mechanism of defense against oxidative stress. Such “hormetic” mechanisms have been likely shaped by Evolution. They are well characterized, for instance, in the case of ionizing radiation (radioactivity), in the sense that limited exposure is beneficial while high exposure is detrimental [45]. In fact, there is a background of radioactivity, mainly due to ^40^K, that is not detrimental to species that have evolved to coexist with such environmental conditions. A significant example of hormetic mechanism in humans was serendipitously discovered as patients of glucose-6-phosphate dehydrogenase (G6PDH) deficiency presented clinical symptoms after intake of antimalaria drug (primaquine) or of fava beans. In fact, in some rural zones the disease was known as favism. When the enzyme is missing the reduced nicotine adenine dinucleotide phosphate (NADPH) required to maintain a functional hemoglobin is not produced, erythrocytes die and hemolysis occurs. In healthy people, intake of fava beans maintains high the levels of G6PDH and, hence, the innate antioxidant mechanism of red blood cells is efficient in keeping significant amounts of reduced glutathione and/or rapidly converting oxidized glutathione into reduced glutathione [26,27,28,29,30] (Figure 2). As mentioned, this mechanism is triggered by the pro-oxidant action of a drug (primaquine) or of a phytochemical (vicine or convicine in fava beans) [29,30] (Table 1). In summary, a pro-oxidant compound wakes-up an antioxidant defense mechanism that is innate in humans. The nutraceutical industry may take advantage of such pro-oxidants to be used at low doses and/or intermittently (a chronic high-dose supplementation is not advisable and would go against the hormetic principle). 

Contrary to what one might think, hormesis is less strange than it seems. Physical training is also an hormetic-like mechanism because sedentary life may lead to oxidative stress due to malfunctioning of cell metabolic events. The oxidative stress in any exercise will be higher in a non-trained individual. Therefore, training in the case of athlets is not only needed to have good scores but to minimize oxidative stress. Trained individuals have all the machinery ready for the exercise and, consequently, oxidative events take place very efficiently and with minimal production of undesired compounds. Furthermore, studies in exercise-trained rats show that diet affects muscle expression of G6PDH [46]. Finally, it has been considered that caloric restriction is an hormetic antiaging mechanism that may increase life span [47], and that antioxidant phytochemicals may be neuroprotective by hormetic processes that involve the engagement of “vitagenes” [48] (Figure 2). The word was coined to highlight those genes that are involved in repair and maintenance processes; Rattan SI (1998) wrote that: the “complex network of the so-called longevity assurance processes is composed of several genes, which may be called vitagenes” [49]. Genes that encode some heat-shock proteins are considered as vitagenes because temperature is one of the most employed hormetic factors. Heat shock produces upregulation of heat-shock proteins, chaperones that preserve the tridimensional structure of proteins and help newly synthesized proteins to be properly folded [49,50]. 

Another example is the protective role of superoxide radicals on hydrogen peroxide stress [51]. Whereas the excess of superoxide radicals is detrimental for cells, the occasional presence of low levels of these species are beneficial not only for cells but for the whole animal. Rattan SI, already in 1998, demonstrated that mild heat shock retards ageing of human fibroblasts and concluded that “These hormesis-like effects of stress-induced defense processes can be useful to elucidate the role of maintenance and repair mechanisms in ageing” [50]. A similar procedure has recently shown that neuronal survival is enhanced. Mild stress applied to primary cultures of cortical neurons decreases the chance of deposition of lipofuscin granules and of pathological aggregates (neurofibrillary tangles or senile plaques) [52]. Mild oxidative stress achieved in worms by downregulating the electron transport chain, affords protection against age-related proteostasis collapse and restores the heat-shock response [53]. At moderate doses, green tea polyphenols reduce the pathological signs in an experimental rodent model of colitis, while unwanted side-effects appear at high doses [54]. 

Glycation is a very well-known phenomenon but difficult to address from a technological point of view. Part of the reason is the structural diversity of sugars coupled to proteins and/or lipids. The simplest example is, perhaps, the so-called advanced glycation endproducts (AGEs) constituted by a sugar and a protein that are formed in the absence of any catalyst. Formation occurs in any condition but it is enriched when sugar blood levels rise (e.g., in diabetes). They are supposedly involved in a variety of diseases from hypertension to neurodegeneration; however, some evidence indicates that AGEs may be protective. For example, ischemic preconditioning is a mechanism of protecting heart from occlusive cardiovascular disease. Murry et al. (1986) showed that cardioprotection can be achieved by short cycles of ischemia-reperfusion [55]. It seems that AGEs formation in these circumstances is part of the protection mechanism (see [56,57]). Hence, it is suggested that moderate stress by AGEs may potentiate innate defense mechanisms in various illnesses [57]. Therefore, protein glycation may be beneficial or detrimental, thus constituting a hormetic response that may be, likely, modulated by plant-derived products. Indeed, this is an attractive field of research.

It has been shown in an invertebrate animal model, *Caenorhabditis elegans*, that variation in hormetic effects is genetically determined [58]. Translated to humans, these findings confirm that hormesis is constituted by mechanisms that have been optimized upon evolution. The challenge is to identify plant-derived products able to restore the efficacy of those mechanisms of protection when they become disfuntional by ageing or by disease. A recent report, has shown that ampelopsin, rosmarinic acid and amorfrutin-A are “hormetins” in human skin fibroblasts undergoing senescence. They protect against telomere length reduction and accumulation of 8-OH-deoxyguanosine, an oxidative DNA damage product, while they upregulate heat-shock protein Hsp70 [59].

## 6. Mitochondria-Related Mechanisms of Plant Antioxidant/Nutraceutical Action

It is well known that polyphenols and other bioactive compounds derived from plants may act on mitochondria. As an example, isoflavones, trans-resveratrol and resveratrol analogues, activate peroxisome proliferator—activated receptor γ (PPARγ) coactivator-1β (PGC-1β) leading to increase the levels of medium-chain acyl-CoA dehydrogenase, a mitochondrial enzyme that participates in lipid metabolism, in a transgenic model of PGC-1β overexpression [60]. The mode of action of these phytochemicals is unclear as PPARγ is a nuclear receptor and the upregulated enzyme is mitochondrial. Daidzein and genistein, two widely-studied phytoflavonoids, protect cerebellum granule neurons from apoptosis by interacting with the mitochondria. In conditions of induced apoptosis, the two compounds “prevented the impairment of glucose oxidation and mitochondrial coupling, reduced cytochrome C release, and prevented both impairment of the adenine nucleotide translocator and opening of the mitochondrial permeability transition pore” [61]. The underlying molecular mechanisms of these effects are not known and the relevant question is whether some pathway or another REDOX reactions are involved. The finding, by Atlante and co-workers (2010), that superoxide compounds decreased the level of reactive oxygen species in conditions of apoptosis is not a proof that anti-apoptotic action is due to any intrinsic antioxidant power of daidzein and genistein.

Plants have chloroplasts and mitochondria both of which are involved in REDOX reactions. Therefore, some of components of these plant organelles, which upon oral intake reach significant levels in blood, may arrive to the mitochondria of mammalian cells and participate in electron transport, or in oxidative phosphorylation or both. Refilling the metabolites in mitochondria by plant-derived products may result in preventing oxidative stress (Figure 2). Conceptually, this antioxidant indirect mechanism is similar to providing supplements to increase the concentration of mammalian cell molecules, the difference is that plant-specific components may directly participate in mammalian mitochondrial REDOX pathways. There is evidence of such possibility. On the one hand, there are reasons to believe that plant-specific mitochondrial components could readily act as mediators of the electron transport events in mammalian cells. On the other hand, it has been shown that polyphenols may act in the mitochondrial machinery independently of reactive oxygen species scavenging; the authors of this study also indicate that “certain polyphenols affect mitochondrial electron transport chain and ATP synthesis” [62]. Thermogenesis and mitochondrial biogenesis are other processes regulated by polyphenols, at least in vitro [63]. However, and despite suggestive data [62,64], it is unlikely that plant polyphenols may act as real anti-cancer compounds by releasing cytochrome C and unrolling apoptosis in tumor cells.

Hints on the direct engagement of endogenous compounds in electron transport chain are many-fold, but two properties are required: (i) the compound reaches the mitochondria, and (ii) the compound has a reduction potential (εº’) between that of the global semi-reaction—soxygen to water and NAD^+^ to NADH; standard reduction potentials in cellular conditions of pH and ion composition (εº’) are, respectively, 0.81 and −0.32 Volts. As a reference the εº’ of ubiquinone to ubiquinol semi-reaction is 0.05 Volts. Since an exhaustive review is not yet possible due to the need of more reports on the subject, a couple of examples will be given. The first is related to quercetin, one of the most studied plant antioxidant. Quercetin is a flavonoid included in human diet because it is present, among other vegetable products, in apples, oranges, lemons, onions, tomatoes and broccoli. Red wine, St John’s wort (*Hypericum perforatum*), or *Ginkgo biloba* are also sources of this compound which, in addition, is sold as a dietary supplement. A very recent study using astrocytes deficient in methyl-CpG-binding protein 2 transcription factor showed that quercetin rescued the reduced activity of mitochondrial respiratory chain complex-II and complex-III [65]. Another recent study demonstrated benefits for mitochondrial function of kolaviron, a biflavonoid found in the seeds of *Garcinia kola* [66]. Kolaviron is able to reverse mitochondrial electron transport chain dysfunction after brain ischemia/reperfusion injury. In this sense, authors conclude that “kolaviron … is a promising candidate for drug development against stroke” [67]. What remains to be established are the details of the mode of action of kolaviron, for instance, which is the targeted mitochondrial complex: I, II, III or IV. 

## 7. Conclusions

Plant antioxidant action for the benefit of human health/well-being may occur by a variety of mechanisms. As in the case of therapeutic drugs, industry has to address the mode of action of “antioxidants”. Direct mechanisms include quick REDOX reactions occurring locally but, also, the action of a given compound that after blood-mediated distribution into a target tissue participates in an enzyme-catalyzed REDOX reaction.

Indirect mechanisms, more difficult to measure but with significant potential to help in development of the industry of plant supplements and nutraceuticals, include actions such as boosting innate mechanisms of detoxification. Hormetic processes, also indirect, should be considered since plant derivatives may provide pro-oxidants able to upregulate the expression of enzymes of innate detox pathways or, alternatively, regulators of the expression of vitagenes (as defined in [48]). 

Finally, it is likely that molecules present in chloroplast and mitochondria of plant cells (and therefore in plant extracts) may reach the mitochondria of mammalian cells to make electron transport and oxidative phosphorylation more efficient. One of the causes of oxidative stress is malfunctioning of mitochondria due to a disease and/or to aging. 

Nutraceutical industry, focusing more on mechanisms, could select the best candidates from the myriad of plant-derived molecules that can be divided as more beneficious or as detrimental (or less beneficious). Furthermore, the industry must consider that, often, in vitro measured antioxidant power does not correlate with antioxidant action at the physiological level. 

## Figures and Tables

**Figure 1 antioxidants-08-00373-f001:**
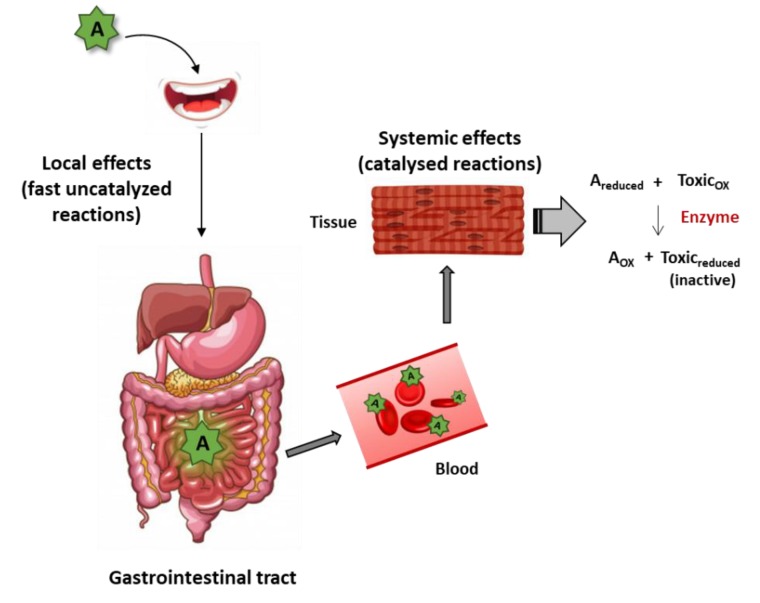
Scheme of direct antioxidant mechanisms including local effects by fast uncatalyzed reactions and systemic effects due to catalyzed reactions. A—antioxidant, A_ox_—oxidized antioxidant, A_reduced_—reduced antioxidant, Toxic_ox_—oxidized toxic, Toxic_reduced_—reduced toxic (non-toxic in reduced form).

**Figure 2 antioxidants-08-00373-f002:**
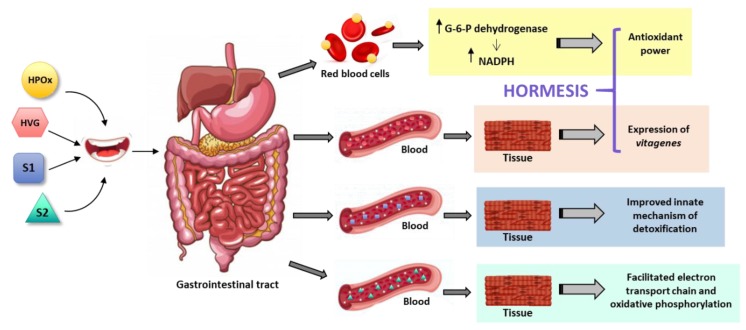
Scheme of indirect antioxidant mechanisms including hormetic actions exerted by pro-oxidants or vitagene expression regulators, and the use of supplements that improves innate mechanisms of detoxification or facilitates mitochondrial function. HPOX—hormetic pro-oxidant, HVG—vitagene-enhancing hormetic compound, S1—supplement type 1, S2—supplement type 2.

**Table 1 antioxidants-08-00373-t001:** Sources, mechanism of action and effects of dietary phytochemicals.

Phytochemicals	Sources	Mechanism of Action	Effects	References
Fructose	Cultivated fruits or sweeteners	Increases in plasma triglycerides, insulin resistance, high blood pressure, etc.	Fatty liver, insulin resistance, dyslipidemia, etc.	[8,9,12,13,14,15,16,17,18,19,20]
Coenzyme Q10	Oranges, spinach, broccoli, soybeans, nuts, sesame seeds, etc.	Correct function of the electron chain transport in mitochondria	Improvement in orthostatic hypotension, renal alterations in type II diabetes	[21,22,23,24]
Lipoic acid	Potatoes, spinach, broccoli, carrots, tomatoes, rice bran, etc.	Correct function of different enzymatic systems	To combat oxidative stress (by mechanisms not known)	[25]
Vicine and convicine	Fava beans	“Hormetic” maintenance of high levels of glucose-6-phosphate dehydrogenase	Maintenance hemoglobin in a functional state and the innate antioxidant mechanism of red blood cells	[26,27,28,29,30]

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
