# Peer review of "Hormetic and Mitochondria-Related Mechanisms of Antioxidant Action of Phytochemicals"

_antioxidants, 2019, doi:10.3390/antiox8090373_

Round 1

Reviewer 1 Report

This review aims at discussing the different mechanisms that regulate antioxidant systems. The authors, assert that there are indirect mechanisms such as hormetic mechanisms. moreover, they cite evidence that about plant-derived compounds may have a direct role in anti-oxidant or pro-oxidant events taking place in mitochondria. I think that the review is not interesting, it is lacking in many parts and there are many points that should be investigated. In particular:

I suggest that the studies cited by the authors to support their thesis should be summarized in a table. In particular, I refer to the studies that the authors cite in paragraph 2, lines106-127.

Page 4 line 153: The authors write “bad” polyphenols. Are there examples in the literature? Please, give some examples.

Paragraph 3: The concept of “nutraceutical substance” is not deepened. I believe that it is useful and necessary to introduce the definition before discussing the mechanisms and potential mechanisms of action of a nutraceutical.

Page 4 line 160: the sentence “if cells have all the required components and the mitochondria are healthy, oxidative stress is negligible” is not correct. The oxidative stress is a pathological condition that is always dangerous for cells and tissues (e.g. see Pham-Huy LA, et al. Int J Biomed Sci. 2008), I suggest to review and rewrite this concept.

Finally, I suggest that the molecules, mechanisms and pathological contest that the authors discuss in paragraph 4, lines167-228 should be summarized in a table.

Author Response

This review aims at discussing the different mechanisms that regulate antioxidant systems. The authors, assert that there are indirect mechanisms such as hormetic mechanisms. Moreover, they cite evidence that about plant-derived compounds may have a direct role in anti-oxidant or pro-oxidant events taking place in mitochondria. I think that the review is not interesting, it is lacking in many parts and there are many points that should be investigated. In particular:

I suggest that the studies cited by the authors to support their thesis should be summarized in a table. In particular, I refer to the studies that the authors cite in paragraph 2, lines106-127.

Answer: we appreciate the suggestion that has been taken into account and a table has been added to the revised version of the manuscript.

Page 4 line 153: The authors write “bad” polyphenols. Are there examples in the literature? Please, give some examples.

Answer: we appreciate raising this issue. In fact, it seems as if all polyphenols are “good” and this would be quite unique. True that recent reports do not raise this issue (perhaps we have overlooked some paper(s)) but really there are polyphenols that are “bad”. Examples are now provided in the revised version of the manuscript.

Paragraph 3: The concept of “nutraceutical substance” is not deepened. I believe that it is useful and necessary to introduce the definition before discussing the mechanisms and potential mechanisms of action of a nutraceutical.

Answer: this good suggestion has been taken into account and the definition of nutraceutical substance has been added, accordingly, to the revised version of the manuscript.

Page 4 line 160: the sentence “if cells have all the required components and the mitochondria are healthy, oxidative stress is negligible” is not correct. The oxidative stress is a pathological condition that is always dangerous for cells and tissues (e.g. see Pham-Huy LA, et al. Int J Biomed Sci. 2008), I suggest to review and rewrite this concept.

Answer: this good suggestion has been taken into account and the sentence has been rewritten in the revised version of the manuscript.

Finally, I suggest that the molecules, mechanisms and pathological contest that the authors discuss in paragraph 4, lines167-228 should be summarized in a table.

Answer: we appreciate the suggestion that has been taken into account and a table has been added to the revised version of the manuscript.

Observation: major changes are highlighted in yellow. Grammatical corrections are not highlighted.

Reviewer 2 Report

The review entitled “Hormetic and Mitochondria-related Mechanisms of Antioxidant Action of Phytochemicals” proposed by Franco et al. deals with an interesting and very timely topic as the antioxidant action of phytochemicals.

General comment - While the authors pay much attention in introducing the themes, these themes did not find in-depth analysis within the manuscript. In fact, despite the idea of hormetic effect suggested is challenging, unfortunately, both the “Mechanisms” and mitochondrial role reported in the title have been only touched on. Also the cited literature appears incomplete and only partially relevant to the theme. For this reasons the manuscript in its present form appears, in the opinion of this reviewer, much more as an “educational” editorial than a scientific review on antioxidant action of phytochemicals.   

Some specific point/suggestions are listed below:

Title: “hormetic” as well as “mitochondria-related mechanism” proposed in the title did not find enough room in the manuscript, this resulting in a mismatch between what expected to read and what reported in the manuscript; Section 2 deals with fructose which cannot be considered as a suitable example of antioxidant molecule, as also recognized by both the authors and literature (besides cited references # 20,21 see also García-Arroyo et al Free Radical Biology and Medicine 141(2019):182-191). No information about mono-disacccharides content is provided in Ref # 10 (L105) in which isoflavones and resveratrols levels were measured, this latter aspect appearing more relevant for the aim of this review, but, unfortunalely, not discussed in the manuscript. Section 3 did not report in detail any direct mechanism of plant antioxidant/nutraceutical action: except for ref #7 no literature evidences were provided about phytochemicals and involved pathways.   Again in section 4, no mechanism is described, and pathways involved are only mentioned without a detailed analysis of literature. In addition, both Q10 and lipoic acid can be found in remarkable content not only in plant but also in red meats and liver, thus they represent “unsuitable” (or at least uncommon) examples of phytochemicals. P(ages) 4-5 L(ines) 190-209: this aspect has been already reported by the same authors in the very recent review “Antioxidant Defense Mechanisms in Erythrocytes and in the Central Nervous System” Antioxidants 2019, 8, 46. P 4 L 210-214: the aspect shortly mentioned in L. 210-212 should be one of the main themes of the present review (as suggested by the title and abstract), as well as what quoted in L.227-228, but no further details were provided. Section 5: one could argue with authors that “an exhaustive review is out if the scope of the present review”, however, being the aim of the paper to deal with mitochondrial-related mechanism and phytochemical action, such argumentation cannot be limited to a “couple of examples” (L.252-263). A more detailed analysis of literature should be provided. Both Figs 1 and 2 did not show any relevant information, thus they did not add much to the paper.

Minors:

Abstract (L26): “sable” ? Ref # 8 needs further details as the source title (book, journal…), volume, pages etc… 39: C. elegans in italics

Author Response

The review entitled “Hormetic and Mitochondria-related Mechanisms of Antioxidant Action of Phytochemicals” proposed by Franco et al. deals with an interesting and very timely topic as the antioxidant action of phytochemicals.

Answer: thanks for the positive comment.

General comment - While the authors pay much attention in introducing the themes, these themes did not find in-depth analysis within the manuscript. In fact, despite the idea of hormetic effect suggested is challenging, unfortunately, both the “Mechanisms” and mitochondrial role reported in the title have been only touched on. Also the cited literature appears incomplete and only partially relevant to the theme. For this reasons the manuscript in its present form appears, in the opinion of this reviewer, much more as an “educational” editorial than a scientific review on antioxidant action of phytochemicals.  

Answer: we appreciate really the comment and we understand the feeling. However, we think that hormesis is virtually unknown in the antioxidant field. In addition, this paper is intended to appear in the special issue of the Antioxidants entitled: “Plant Antioxidant for Application in Food and Nutraceutical Industries”. Accordingly, we wrote the paper more as “educational” than scientific and, apparently, we have succeeded. Albeit the paper may appear as educational, it is based on relevant scientific reports.

Not only hormesis is little known but, apart from the “fava beans” example, there are few studies in the plant antioxidant field to write a review. Actually, we have modified the sentence that stated that a review “is out of the scope” as there is no enough data in this field for a review. This last sentence does not mean that we do not now consider our paper a review, but a review on mechanisms.

Some specific point/suggestions are listed below:

Title: “hormetic” as well as “mitochondria-related mechanism” proposed in the title did not find enough room in the manuscript, this resulting in a mismatch between what expected to read and what reported in the manuscript

Answer: our paper is a review on mechanisms and we agree in that in the examples given, more information may be added so we have provided more information in the revised version of the article. Again, more examples are not necessary, in our opinion, (not many good Plant antioxidant-related examples) as more examples may blur the message.

Section 2 deals with fructose which cannot be considered as a suitable example of antioxidant molecule, as also recognized by both the authors and literature (besides cited references # 20,21 see also García-Arroyo et al Free Radical Biology and Medicine 141(2019):182-191). No information about mono-disacccharides content is provided in Ref # 10 (L105) in which isoflavones and resveratrols levels were measured, this latter aspect appearing more relevant for the aim of this review, but, unfortunately, not discussed in the manuscript.

Answer: we agree in that section 2 could be improved and we have split it into two. We have added the indicated (very recent) reference. In fact, we appreciate detecting the mistake in the reference #10 as the correct one (old one replaced by this one) is:

Ma, B.; Chen, J.; Zheng, H.; Fang, T.; Ogutu, C.; Li, S.; Han, Y.; Wu, B. Comparative assessment of sugar and malic acid composition in cultivated and wild apples. Food Chem. 2015, 172, 86–91.

We do not understand how levels of isoflavones and resveratrol may improve our paper.

Section 3 did not report in detail any direct mechanism of plant antioxidant/nutraceutical action: except for ref #7 no literature evidences were provided about phytochemicals and involved pathways.   Again in section 4, no mechanism is described, and pathways involved are only mentioned without a detailed analysis of literature. In addition, both Q10 and lipoic acid can be found in remarkable content not only in plant but also in red meats and liver, thus they represent “unsuitable” (or at least uncommon) examples of phytochemicals.

Answer: of course, coenzyme Q10 and lipoic acid are in animals and animal-derived products are in our diet. Why coenzyme Q10 and lipoic acid supplements are consumed and actually are in “fashion”?  Coenzyme Q10 and lipoic acid are good examples of vehicles with indirect antioxidant action and this is important to understand the “refuelling mechanism”. Within a “Plant Antioxidant for Application in Food and Nutraceutical Industries” topic the examples are, in our opinion, appropriate.  Going to Amazon we find:

Lipoic acid described as Alpha-Lipoic Acid, Non-GMO, Gluten Free, Vegan, Soy Free, helps Maintain Blood Sugar Levels Not clear the source; perhaps using a synthetic product. Googling “sources of Q10 supplements” we find animal and those plant-derived sources:

Vegetables: spinach, cauliflower and broccoli. Fruit: oranges and strawberries. Legumes: soybeans, lentils and peanuts. Nuts and seeds: sesame seeds and pistachios.

But improving the search adding “industrial Q10” we find a paper Parmar et al., Crit Rev Biotechnol 2015 that states: “Although, microbial production is the major industrial source of CoQ10 but due to low yield and high production cost, other cost-effective and alternative sources need to be explored. Plants, being photosynthetic, producing high biomass and the engineering of pathways for producing CoQ10 directly in food crops will eliminate the additional step for purification and thus could be used as an ideal and cost-effective alternative”

In summary, we find those examples very appropriate for an article within the Plant Antioxidant for Application in Food and Nutraceutical Industries” topic.

P(ages) 4-5 L(ines) 190-209: this aspect has been already reported by the same authors in the very recent review “Antioxidant Defense Mechanisms in Erythrocytes and in the Central Nervous System” Antioxidants 2019, 8, 46. P 4 L 210-214: the aspect shortly mentioned in L. 210-212 should be one of the main themes of the present review (as suggested by the title and abstract), as well as what quoted in L.227-228, but no further details were provided.

Answer: as earlier mentioned, hormesis is not well known in this field. Actually, the indicated example was not considered hormetic until a reviewer of the paper convinced us. We are now convinced and we use the same example as it is “old/classical” and very illustrative/informative. But we appreciate the comment and we agree in that more information can be added in the places indicated by the reviewer. This issue has been considered in the revised version of the paper.

Section 5: one could argue with authors that “an exhaustive review is out if the scope of the present review”, however, being the aim of the paper to deal with mitochondrial-related mechanism and phytochemical action, such argumentation cannot be limited to a “couple of examples” (L.252-263). A more detailed analysis of literature should be provided.

Answer: as earlier mentioned, we think that examples may include more information but providing an excess of examples would blur the messages (the various messages as the paper focuses on diverse mechanisms). In addition, we think that there are few plant-antioxidant-related examples; if we have missed a seminal paper/example we would appreciate to know the details to include it in our manuscript.

Both Figs 1 and 2 did not show any relevant information, thus they did not add much to the paper.

Answer: perhaps this is the summary of the disagreement in the views of the reviewer and our views on how to construct this paper. We selected this option and, honestly, we think that as it is (but improved after addressing the issues of the three reviewers), it is sound and timely. Concerning figures, we cannot agree with the reviewer. We have though in modifying some and/or adding another one but we always end up in the same “place”: to our knowledge there are no similar figures in the literature. Furthermore, they are very illustrative, in our opinion, and we think that the field will appreciate them. If there are better illustrations in this specific subject and for this specific special issue, we will appreciate to know where. As per the request of another reviewer we have incorporated the reference to the figures within the text.

Minors:

Abstract (L26): “sable”?

Answer: thanks for noticing the mistake, which is considered in the revised version of the manuscript.

Ref # 8 needs further details as the source title (book, journal…), volume, pages etc…

Answer: we appreciate the comment, which is taken into account in the revised version of the manuscript. This reference has been corrected. Thanks for noticing the mistake.

39: C. elegans in italics

Answer: we appreciate the comment, which is considered in the revised version of the manuscript. Thanks for noticing the mistake.

Observation: major changes are highlighted in yellow. Grammatical corrections are not highlighted.

Reviewer 3 Report

- The main issue about this review paper is that in the title the authors try to describe the hormetic and mitochondria-related mechanisms of antioxidant action. So the introduction should justify more properly the reasons and the background to make this review. Also, the conclusions should give a more precise answer to the aims. - The authors may a difference between direct and indirect Mechanisms of Plant Antioxidant/Nutraceutical Action. It is no clear the reason to differentiate both aspects and the influence in the hormesis. - Fig 1 and 2 should be included and described in the main text, not in the conclusions - Page 4 line 150 specific enzyme able to catalyze de REDOX reaction, change de to the - The authors write: “This paper aims at discussing mechanisms by which “antioxidants” from plants may exert a REDOX-related beneficial action”. Why they describe in the first part that the Antioxidants Are Not Needed for Animal Life. This issue reduces the interest in the main paper´s goal. Maybe it could be nice to describe the hormetic and mitochondria-related mechanisms of antioxidants action that can be changed by the phytochemical use.

Author Response

The main issue about this review paper is that in the title the authors try to describe the hormetic and mitochondria-related mechanisms of antioxidant action. So the introduction should justify more properly the reasons and the background to make this review. Also, the conclusions should give a more precise answer to the aims.

Answer: thanks for the positive comment.

The authors may a difference between direct and indirect Mechanisms of Plant Antioxidant/Nutraceutical Action. It is no clear the reason to differentiate both aspects and the influence in the hormesis.

Answer: thanks for the positive comment. We have rewritten some parts of the paper to avoid confusion.

Fig 1 and 2 should be included and described in the main text, not in the conclusions

Answer: this good suggestion has been taken into account, and figures 1 and 2 have been included in the main text.

Page 4 line 150 specific enzyme able to catalyze de REDOX reaction, change de to the

Answer: Thanks for noticing this mistake.

The authors write: “This paper aims at discussing mechanisms by which “antioxidants” from plants may exert a REDOX-related beneficial action”. Why they describe in the first part that the Antioxidants Are Not Needed for Animal Life. This issue reduces the interest in the main paper´s goal. Maybe it could be nice to describe the hormetic and mitochondria-related mechanisms of antioxidants action that can be changed by the phytochemical use

Answer: we appreciate this consideration. In our opinion, the way of presenting the issue is attractive (but we have tried to improve it in the revised version) as antioxidants, indeed, are not needed until they are needed (it seems an oxymoron and this is perhaps the “issue”). This “oxymoron” is a key factor as there is not expected that a baby takes an antioxidant for survival; nobody even thinks of it. Then, the key issue is that at some point we look for antioxidants for a longer and a healthier life. We have modified the way to enter into this seminal (to us) view to try to make it both more comprehensible and attractive.

Observation: major changes are highlighted in yellow. Grammatical corrections are not highlighted.

Round 2

Reviewer 1 Report

Thank you for the ansewers. I have no further questions.

Author Response

Thanks for your help in improving the paper

Reviewer 2 Report

The revised version of the manuscript by Franco et al. appears partially improved by reporting a deeper insight into some of the topics they propose. A clearer explanation of vitagenes (and a couple of examples) was reported. Other references were added, as well as further “hormetic” examples were provided, thus increasing the scientific significance of the paper. Despite that, some of the criticisms raised in the first review are still present. In particular section 6 results to be improved. In this regard, I can suggest a couple of papers (but there is a huge number) worthy to be quoted in this section: the direct influence of genistein, daidzein on mitochondria of apoptotic neuronal cells was studied in Atlante et al., Biochemical Pharmacology 79(2010):758-767; whereas an “indirect” effect exerted by the same compounds was very recently reported by Uchitomi et al., Biochemistry and Biophysics Reports 17(2019): 51-55. In my opinion, the presence of these papers (but also others) would strengthen this section.

Moreover, in spite of what stated in both the manuscript and the reply letter, the choice of reporting, almost exclusively, CoQ10 and lipoic acid as examples of antioxidant phytochemical remains atypical, mostly in the light of the lack of a dissertation on polyphenols and flavonoids (see table I). Really do the authors think that isoflavones and resveratrol content in food cannot be of interest in their review (as they state in their reply letter)? Notice that by searching for “resveratrol and mitochondria” (as an example) on Scopus more than 1,300 papers were found. Thus, a mention on the role of plant phenols cannot be neglected in the sections 2-3.

Also the choice of deleting Lee’s and Kwon’s papers (old refs 24-25) in the revised version appears strange since both described/proposed mechanisms of action for some flavonols.

Again in section 2 (Line 57-65): I cannot agree with what reported in these sentences, and especially where they stated that “…antioxidants are not required for a child’s healthy life”, since Vitamin A and mostly Vitamin E act as antioxidant (with the latter known to be the most powerful). Both of them are transferred to the offspring during lactation (see Debier and Larondelle. British Jounal of Nutrition 93(2005):153-74) and also commercial infant milk formulation contains vitamins (including A and E). I think this part should be revised.

Line 92-95: lactose in bloodstream as a result of fructose metabolism??? Maybe authors refer to lactate!!

Although the new position for fig. 1 appears more logical in the educational style used by the authors, I still feel that this scheme did not add more to the paper; moreover, it should be noted in this regard that antioxidant bioavailability, absorption process and cellular uptake were completely neglected in this very simplified version. On the contrary, these items are among the most investigated, but elusive, aspects in the research in this field (in this regard Vincenzo Fogliano’s papers can be of help). Thus, some considerations/comments about tissue and cellular uptake processes should be provided in section 4.  

Minors:

Line 330: “?” in section 6 title 52: “C. elegans” was not in italics

Reviewer 3 Report

Now it can be accepted

Author Response

Thanks for your help in improving the manuscript